# Personalised Approach to Diagnosing and Managing Ischemic Stroke with a Plasma-Soluble Urokinase-Type Plasminogen Activator Receptor

**DOI:** 10.3390/jpm12030457

**Published:** 2022-03-14

**Authors:** Katarzyna Śmiłowska, Marek Śmiłowski, Robert Partyka, Danuta Kokocińska, Przemysław Jałowiecki

**Affiliations:** 1Department of Emergency Medicine, Faculty of Medical Sciences, Medical University of Silesia, 40-055 Katowice, Poland; robertpartyka@op.pl (R.P.); dkokocinska@op.pl (D.K.); olaf@pro.onet.pl (P.J.); 2Department of Neurology, 5th Regional Hospital in Sosnowiec, Plac Medyków 1, 41-200 Sosnowiec, Poland; 3Department of Hematology and Bone Marrow Transplantation, Medical University of Silesia, 40-055 Katowice, Poland; marek.smilowski2@gmail.com

**Keywords:** stroke, ischemia, biomarkers, mortality risk

## Abstract

Background: The increasing incidence of ischemic stroke has led to the search for a novel biomarker to predict the course of disease and the risk of mortality. Recently, the role of the soluble urokinase plasminogen activator receptor (suPAR) as a biomarker and indicator of immune system activation has been widely examined. Therefore, the aim of the current study was to assess the dynamics of changes in serum levels of suPAR in ischemic stroke and to evaluate the prognostic value of suPAR in determining mortality risk. Methods: Eighty patients from the Department of Neurology, diagnosed with ischemic stroke, were enrolled in the study. Residual blood was obtained from all the patients on the first, third and seventh days after their ischemic stroke and the concentrations of suPAR and C-reactive protein (CRP), as well as the number of leukocytes and National Institute of Health’s Stroke Scale (NIHSS) scores, were evaluated. Results: On the first day of ischemic stroke, the average suPAR concentration was 6.55 ng/mL; on the third day, it was 8.29 ng/mL; on the seventh day, it was 9.16 ng/mL. The average CRP concentration on the first day of ischemic stroke was 4.96 mg/L; on the third day, it was 11.76 mg/L; on the seventh day, it was 17.17 mg/L. The number of leukocytes on the first day of ischemic stroke was 7.32 × 10^3^/mm^3^; on the third day, it was 9.27 × 10^3^/mm^3^; on the seventh day, it was 10.41 × 10^3^/mm^3^. Neurological condition, which was assessed via the NIHSS, on the first day of ischemic stroke, was scored at 10.71 points; on the third day, it was scored at 12.34 points; on the seventh day, it was scored at 13.75 points. An increase in the values of all the evaluated parameters on the first, third and seventh days of hospitalisation was observed. The patients with hypertension, ischemic heart disease and type 2 diabetes showed higher suPAR and CRP concentrations at the baseline as well as on subsequent days of hospitalisation. The greatest sensitivity and specificity were characterised by suPAR-3, where a value above 10.5 ng/mL resulted in a significant increase in mortality risk. Moreover, an NIHSS-1 score above 12 points and a CRP-3 concentration above 15.6 mg/L significantly increased the risk of death in the course of the disease. Conclusions: The plasma suPAR concentration after ischemic stroke is strongly related to the patient’s clinical status, with a higher concentration on the first and third days of stroke resulting in a poorer prognosis at a later stage of treatment. Therefore, assessing the concentration of this parameter has important prognostic value.

## 1. Introduction

Clinically, stroke is defined as a focal or global cerebral dysfunction, of a vascular origin, that occurs suddenly and lasts at least 24 h [1]. Over 12 to 15 million people worldwide are afflicted by this disease annually [2,3]. Approximately 15% of patients die within one month after having a stroke, and only 10% present a complete withdrawal of neurological deficits. The risk of recurrent ischemic stroke is 5 to 25% annually and is higher in the first weeks following the stroke [4,5,6]. In general, 25% of the patients present mild neurological symptoms, while 40% experience permanent disability, both motor and cognitive, at a moderate or severe level; another 10% of the patients require continuous nursing care due to neurological disability [7]. This leads to a reduction in daily activities due to varying degrees of neurological deficits [8]. Therefore, the increasing incidence of stroke has prompted researchers to search for new diagnostic biomarkers to enable early disease detection, quick, tailored treatment and, ideally, prognostic value.

Functional impairment from ischemic stroke is the result of not only structural brain damage but also accompanying immune dysregulation [9]. Therefore, the role of inflammatory biomarkers, such as plasma-soluble urokinase-type plasminogen activator receptor (suPAR), C-reactive protein (CRP), procalcitonin (PCT) and white blood cells (WBC), has been strongly argued to predict the course of ischemic stroke and to determine its mortality risk [10]. Increased concentrations of inflammatory markers in the blood occur independent of the infection, and, thus, monitoring these parameters is useful in predicting the course of ischaemic stroke [11].

Urokinase plasminogen activator (uPA) and its receptor (uPAR) play an important role in the pathogenesis of vascular diseases [12]. As a result of the activation of the immune system accompanying a stroke, the concentration of suPAR in body fluids increases with the severity of the immune response [13,14]. So far, evidence is limited regarding whether the increase in suPAR concentration is merely a consequence of the activation of the immune system or also has the potential to enhance immune response [15]. The latter hypothesis is supported by the fact that uPA and its receptor are involved in the conversion of plasminogen to plasmin. Proper blood clotting, conditioned by the dissolution of fibrin at the site of vascular injury, depends on plasminogen activation by a tissue plasminogen activator (t-PA). The uPA–uPAR complex increases plasminogen activity on the cell surface by initiating a proteolysis reaction, which enables cell migration [16]. Moreover, the uPA–uPAR complex has an effect on leukocytes independent of plasminogen activation. The uPA fragments may act as a chemotactin for neutrophils and a mitogen for lymphocytes. Urokinase from the uPA–uPAR complex determines the effective process of adhesion, increasing the interaction with integrins and vitronectin [17]. The interaction of uPAR with β2-integrin increases the migration of inflammatory cells as well as the mobilisation and activation of leukocytes. Moreover, uPAR facilitates migration through plasmin generation on the cell surface (after binding its ligand-uPA) with the subsequent dissolution of the extracellular matrix and plays a role in the mobilisation and activation of leukocytes via interaction with β2-integrins.

Atherosclerotic plaque in the carotid arteries causes an increased concentration of macrophages and leads to the release of uPAR from their surface [18]. Moreover, uPAR plays a role in the pathogenesis of vascular diseases, which is usually accompanied by inflammation [19,20]. Atherosclerotic plaque lesions induce an inflammatory process, which is initiated by low density lipoprotein (LDL) molecules in the subendothelial area of arteries [21]. These molecules induce endothelial cell activation and inflammatory cell recruitment through chemokines and adhesion molecules [22]. The accumulation of inflammatory cells leads to the degeneration of components that stabilise the vascular wall, leading to the weakening of the plaque and increasing the risk of rupture [23]. The degeneration of the extracellular matrix also involves the fibrinolytic system through the activation of plasminogen, which is converted from an inactive proenzyme to plasmin by two activators: tissue plasminogen activator (t-PA) and urokinase plasminogen activator (u-PA). This, in turn, is connected to the cellular uPA receptor. Moreover, the conversion of plasminogen to plasmin leads to fibrin degeneration [24]. A soluble form of defragmented uPAR becomes detectable in body fluids [25]. The plasma concentration of suPAR increases, reflecting the activation of the immune system caused by bacterial and viral infections, sepsis or cancer [26,27].

Ischemic stroke has a two-phase effect on the peripheral immune system. In the initial phase, a generalised inflammatory process results in the massive production of classical inflammatory markers, such as cytokines and chemokines [28]. As a result, the blood–brain barrier enables the infiltration of lymphocytes [29,30,31]. This process leads to secondary brain injury but also plays a protective role as it contributes to the regeneration of nervous tissue.

Damage to the blood–brain barrier results in a local inflammatory reaction that occurs as early as 30 min after the ischemic episode. Neutrophils begin to accumulate in the region of infarction, with the highest concentration occurring between one and three days after ischemia [29]. Neutrophils participate in the inflammatory process by releasing pro-inflammatory mediators [32]. In addition, monocytes are involved in the process, transforming into macrophages in the nervous tissue. They release interleukin-6 (IL-6), which is responsible for immune response within both the central and peripheral nervous systems [33]. Activated T lymphocytes are also involved in the local inflammatory response. Their highest concentration can be observed on the seventh day after the stroke [34]. These lymphocytes—specifically CD4^+^ and CD8^+^ lymphocytes—exert adverse effects by producing interferon-γ and interleukin-17 (IL-17). T-regulatory lymphocytes, in turn, produce interleukin 10 (IL-10), a modulating anti-inflammatory response [35]. Pro-inflammatory cytokines activate local microglial cells and induce the migration of immune cells to the site of ischemia [36].

Stroke-induced immunosuppression (SIIS) occurs after the early inflammatory response [37]. SIIS limits the regeneration of the nervous tissue, worsening the prognosis of and leading to severe complications from the stroke—specifically infections, which occur in up to 65% of patients [38]. The most common infections involve the urinary and respiratory tracts [39]. The increased risk of infection is a result of decreased activation of T lymphocytes and a significant decrease in the number of T and B cells [31]. The activation of the immune system also increases the concentration of inflammatory biomarkers (e.g., CRP, erythrocyte sedimentation rate, white blood cells (WBC), peripheral neutrophils), indicating worse prognosis [40,41,42,43].

We hypothesised that ischemic stroke, which activates the immune system, leads to changes in the concentration of plasma suPAR. Therefore, we sought to accomplish the following objectives: (1) assess the dynamics of changes in plasma suPAR concentrations on the first, third and seventh days after ischemic stroke; (2) conduct a comparative analysis of changes in suPAR concentrations in relation to the neurological status of patients, assessed using the National Institute of Health’s Stroke Scale (NIHSS); (3) assess the correlation between the concentration of suPAR, the concentration of CRP and leukocyte count; (4) assess the prognostic value of suPAR in determining the risk of death among patients with ischemic stroke in comparison with the prognostic value of CRP, the NIHSS and WBC.

## 2. Material and Methods

Eighty patients with ischemic stroke were included in the study in the Department of Neurology of the Regional Hospital in Pszczyna, Poland. The diagnosis was made based on clinical symptoms and neuroimaging studies. Ethical approval for this study was obtained from the Medical University of Silesia in Katowice (no: KNW/0022/KB/85/13). The study was conducted between 2013 and 2017.

The inclusion criteria were as follows: (1) symptoms of stroke up to 24 h prior to admission; (2) >18 years old for both males and females; (3) no other causes of neurological deficits after brain imaging; (4) clinical diagnosis of ischemic stroke. Patients were excluded from participation if they had (1) clinical or laboratory features of infection upon admission to hospital; (2) stroke not ischemic in nature (e.g., haemorrhagic stroke, subarachnoid haemorrhage); (3) other causes of neurological deficits (epileptic seizure with Todd paresis); (4) symptoms of transient ischemic attack (TIA); (5) electrolyte disturbances; (6) cancer diagnosis; (7) stroke located in the posterior cranial fossa.

SuPARnostic kits (ViroGates, Lyngby, Denmark) were used to determine the suPAR concentration. Blood was collected on the first, third and seventh days after ischemic stroke. The material was centrifuged at a rate of 3000 rpm for 10 min. The obtained plasma was frozen at −20 °C. The plasma was then quantitatively analysed using the SuPARnostic ELISA assay (ViroGates, Lyngby, Denmark) to determine the suPAR concentration.

### Statistical Methods

The type of distribution of the examined parameters was evaluated using the Shapiro–Wilk test. A Student’s *t*-test for both related and unrelated variables was used to evaluate mean values whose distribution was close to normal. The statistical analysis assumed a level of significance of *p* < 0.05.

If the distribution differed from the normal distribution, the following procedures were applied accordingly: a Wilcoxon test and a Mann–Whitney U-test. The correspondence of the distribution with the normal distribution was assessed using a chi-square test. For the analysis of non-parametric variables (prevalence), either the non-parametric chi-square independence test or the Fischer precision test was used. Correlations were investigated using the chi-square test. In order to assess the correlation between the examined variables, a significance level of *p* < 0.05 was adopted. To determine the relationship between suPAR, NIHSS, CRP and WBC and the mortality rate, a logistic regression analysis was used, taking into account the risk factors of stroke. The cut-off points for each parameter were determined using area under the curve (AUC) receiver operating characteristic (ROC) analysis, on the basis of which predictive values were obtained for each parameter and a quotient of the likelihood of the occurrence of a given event was determined. In the case of AUC ROC analysis, a significance range of 0.8 to 0.95 was used.

## 3. Results

### 3.1. Study Population

The final analysis included data obtained from 80 patients (37 women and 43 men) who met the inclusion criteria. The demographic characteristics of the study group are presented in Table 1.

### 3.2. Values of the Tested Parameters

The average values of the tested parameters (suPAR, NIHSS, CRP and WBC) are presented in Table 2. On the first day following ischemic stroke, the suPAR (suPAR-1) concentration was 6.55 ng/mL and exceeded the reference values for healthy subjects (4.5 ng/mL). The CRP concentration on the first day following ischemic stroke (CRP-1) was in the upper limit of the normal range (4.96 mg/L; N < 5mg/L), while the number of WBC was 7.32 thousand/mm^3^ and was, therefore, in the normal range (n 9.8 thd/mm^3^).

### 3.3. Assessment of the Change in suPAR, NIHSS, CRP and WBC

Comparing the average suPAR-1 values with those of suPAR-3 and suPAR-7, we observed a statistically significant increase in the suPAR concentration on the third day of hospitalisation (suPAR-3) in comparison with the first day (suPAR-1) (*p* < 0.05) and on the seventh day of hospitalisation (suPAR-7) in comparison with the first and third days (*p* < 0.05).

Based on the analyses, a statistically significant difference was observed in the NIHSS score on the third day of hospitalisation (NIHSS-3) in comparison with the first day of hospitalisation (NIHSS-1) (*p* < 0.05) and on the seventh day of hospitalisation (NIHSS-7) in comparison with the first and third days (*p* < 0.05). A significant increase in the CRP concentration was observed on the third day of hospitalisation (CRP-3) compared to the first day (CRP-1) (*p* < 0.05) and on the seventh day (CRP-7) compared to the first and third days (*p* < 0.05).

The increase in the number of WBC on the third day of hospitalisation (WBC-3) compared to the first day (WBC-1) was also statistically significant (*p* < 0.05), as was the increase in the number of WBC on the seventh day of hospitalisation (WBC-7) compared to the first and third days (*p* < 0.05).

### 3.4. Analysis of suPAR in Correlation to NIHSS

#### 3.4.1. Day I

Statistical analysis revealed a correlation between the suPAR concentration on the first day of hospitalisation (suPAR-1) and neurological status assessed based on the NIHSS (NIHSS-1) (r = 0.48; *p* < 0.05), as well as an average positive correlation between the suPAR-1 concentration and NIHSS-3 (r = 0.47; *p* < 0.05) and NIHSS-7 (r = 0.41; *p* < 0.05).

#### 3.4.2. Day III

On the third day of hospitalisation, a weak positive correlation between the suPAR-3 concentration and NIHSS-3 (r = 0.28; *p* < 0.05) was found.

#### 3.4.3. Day VII

On the seventh day of hospitalisation, a weak positive correlation between the suPAR-7 concentration and NIHSS-7 (r = 0.27; *p* < 0.05) and an average positive correlation between the suPAR-7 concentration and the neurological status of the patient at the same time (NIHSS-7) (r = 0.41; *p* < 0.05) was observed.

### 3.5. The Correlation between Concentrations of suPAR, CRP and WBC

An average positive correlation between the concentration of suPAR-1 and the concentration of CRP-1 (r = 0.43; *p* < 0.05) and a weak positive correlation between the concentrations of suPAR-1 and WBC-7 (r = 0.23; *p* < 0.05) were identified. There was also a weak positive correlation between the concentrations of suPAR-3 and WBC-7 (r = 0.26; *p* < 0.05) as well as between the concentrations of suPAR-7 and WBC-7 (r = 0.38; *p* < 0.05).

There was an average positive correlation between CRP-1 and NIHSS-1 (r = 0.48; *p* < 0.05), between CRP-1 and NIHSS-3 (r = 0.39; *p* < 0.05) and between CRP-1 and NIHSS-7 (r = 0.36; *p* < 0.05).

### 3.6. Impact of Risk Factors on the Assessed Parameters

We analysed the relation between individual risk factors of ischemic stroke, such as hypertension, ischemic heart disease, atrial fibrillation, type 2 diabetes mellitus, smoking and hypercholesterolemia, and their influence on the evaluated parameters.

The patients with arterial hypertension exhibited higher concentrations of suPAR-1, suPAR-3 and suPAR-7, as well as CRP-1, CRP-3 and CRP-7, than the patients without arterial hypertension. Other parameters (NIHSS and WBC) did not differ significantly in the groups of patients with and without hypertension.

In the patients with ischemic heart disease, significant differences were found in both suPAR (suPAR-1, suPAR-3 and suPAR-7) and CRP (CRP-1, CRP-3 and CRP-7) concentrations. There was no correlation between the NIHSS values and the number of WBC in this group of patients.

No differences were found in the parameters after taking into consideration patients with atrial fibrillation.

The patients with type 2 diabetes mellitus showed increased suPAR values in all the time periods (suPAR-1, suPAR-3 and suPAR-7) and in CRP-3 and CRP-7 values in comparison with the patients without diabetes. The remaining parameters did not differ significantly.

Smokers showed higher concentrations of suPAR-3 and suPAR-7, NIHSS-3 and NIHSS-7, as well as CRP-7, compared to non-smokers.

The patients with hypercholesterolemia did not show any differences in the values of the evaluated parameters in comparison with the patients without hypercholesterolemia, except for CRP-7.

### 3.7. Prognostic Value of Assessed Parameters and Mortality Rate

During hospitalisation, 11 patients died (14%). Death occurred on average 17 days after the ischemic episode. All the deaths were related either directly (e.g., brain oedema) or indirectly (e.g., infectious complications) to the stroke. As shown in Table 3, the patients with comorbidities, such as arterial hypertension, ischemic heart disease, type 2 diabetes mellitus and smoking, had a higher mortality risk. Similarly, higher concentrations of suPAR-1, suPAR-3, suPAR-7, CRP-3 and CRP-7, as well as the neurological status of the patient (NIHSS-1, NIHSS-3 and NIHSS-7), were also associated with a higher mortality risk (Table 4).

In contrast, atrial fibrillation and hypercholesterolemia were not associated with mortality risk.

### 3.8. AUC ROC Analysis of the Prognostic Value of suPAR

To determine the risk factors influencing the concentrations of the assessed parameters, a multifactor analysis was conducted to assess their prognostic value in the study group with regard to gender, age, presence of ischemic heart disease, presence of arterial hypertension, presence of type 2 diabetes mellitus and smoking status. The selected group of patients was statistically analysed in order to determine the prognostic value of these parameters in predicting the risk of post-stroke mortality.

Moreover, suPAR-3 had the highest sensitivity and specificity, with a cut-off point of 12.7 ng/mL, indicating an increased risk of post-stroke mortality. A concentration of suPAR-1 exceeding 8.4 ng/mL was associated with a higher risk of death. NIHSS-1 (≥14 points) and CRP-3 (concentration above 14.6 mg/L) were correlated with a significant increase in mortality risk (Table 5).

## 4. Discussion

To our knowledge, this was the first study to examine the prognostic value of suPAR concerning ischemic stroke. The majority of the previous studies have focused on suPAR concentrations with regard to ischemic heart disease, cancer, infections or type 2 diabetes mellitus.

The suPAR concentration increased on the first day following ischemic stroke, with further increases in the subsequent days. The increases in CRP and WBC were observed later, starting on the third and seventh days after the stroke, respectively. Folyovich et al. made a similar observation, reporting that the suPAR concentrations increased immediately after the stroke and continued to increase for the next seven days, while CRP and WBC increased only on the seventh day after the stroke [31].

Based on our results, a suPAR-1 concentration above 7.64 ng/mL increases the mortality risk almost threefold, while a suPAR-3 concentration above 10.49 ng/mL increases the mortality risk by up to eightfold. We also found that an NIHSS score of >12 on the first day after a stroke increases the mortality risk threefold. At the same time, a CRP-3 concentration >15.6 mg/L was associated with a fivefold increase in the mortality risk. The prognostic value of WBC was found to be significant only on the seventh day following the stroke, doubling the mortality risk. The AUC in the ROC analysis for suPAR-3 was the largest, amounting to 0.89, and this parameter had the highest prognostic value on the third day after the stroke.

The concentration of suPAR, similar to that of CRP, plays an important role in predicting the course of disease and the risk of mortality among patients with life-threatening conditions. This argument applies not only to stroke but also to cancer and infectious diseases. The relationship between suPAR levels and mortality risk has been observed in malaria, tuberculosis, HIV infections, urinary tract infections and bacterial meningitis, as well as in neoplastic diseases, such as colorectal cancer, ovarian cancer and multiple myeloma [44,45,46,47,48,49,50]. Increased suPAR concentrations indicated a poor prognosis in bacteraemia [51,52], and its prognostic role was also demonstrated in patients with acute respiratory distress syndrome (ARDS)—i.e., the suPAR concentration correlated with the severity of ARDS [53]. In HIV infections, a higher concentration of suPAR was observed in patients with a lower number of CD4^+^ lymphocytes, a higher viral load and greater AIDS-related mortality. Patients diagnosed with fatal sepsis also had significantly higher suPAR concentrations than those with non-fatal sepsis. As a single biomarker, suPAR had a higher prognostic value than CRP and PCT, increasing even further when analysed in combination with the above biomarkers. Increased concentrations of suPAR were associated with an increased likelihood of being transferred to the intensive care unit [54,55]. Moreover, in this case, the suPAR concentrations correlated with the assessment scales typically used in intensive care units, such as APACHE-II, SAPS-II or SOFA. Donadello et al. demonstrated that, in sepsis, a suPAR concentration of >5.5 ng/mL has a high prognostic value, exceeding that of both CRP and PCT [56]. Donadello et al. pointed out that sepsis generates many reactions from both inflammatory and anti-inflammatory mediators, activating cellular and also humoral responses. Likewise, in a study by Kofoed et al., the prognostic value of suPAR was compared with that of CRP and PCT in patients with sepsis [55,56]. The prognostic value of suPAR was higher than that of other markers and was comparable to the SOFA or SAPS II score; further, combining suPAR with age generated a better prognostic result than the SAPS II score alone.

### 4.1. Markers of Inflammation in Stroke

Emsley et al. reported that the CRP concentrations may already be elevated on the day of the stroke, with the highest concentrations occurring 5 to 7 days thereafter and remaining stable for the subsequent three months [57]. These results are similar to those reported in our study, although the time points were slightly different. Additionally, the dynamics of changes in leukocyte counts reported by Emsley et al. were slightly different than those we reported. On the first day of measurement, the number of WBC did not exceed the reference values (9.8 thousand/mm^3^), although they did significantly differ from those in the control group (6.2 thousand/mm^3^). Emsley et al. also observed an increase in the number of WBC above the normal count between five and seven days following stroke and, as in the case of CRP, these values were maintained for three months thereafter. These observations were made after the exclusion of the infectious factor on admission to the hospital. According to the authors, two mechanisms may be the cause of the increase in CRP and WBC. On the one hand, strokes cause the rapid activation of the immune system, which persists in the first days following the stroke; on the other hand, it is possible that increased levels of inflammatory parameters may result from a “low-activity” inflammatory process preceding the stroke, one that develops asymptomatically in the body, contributing to the vascular incident.

The above hypothesis could be confirmed by observations made by Rost et al., which also indicated the contribution of a “low-activity” inflammatory process to the aetiology of stroke and transient cerebral ischemia [58]. During 12 to 14 years of observation, the authors found that CRP concentrations in the upper range of normal values increased the risk of ischemic brain incidents threefold in women and twofold in men, including after considering the co-occurrence of risk factors such as smoking, cholesterol, systolic blood pressure or type 2 diabetes mellitus. The authors stated that the relationship between CRP concentrations and risk of stroke was linear and that the high values of this relationship can be treated as an independent risk factor for stroke. A similar relationship has also been demonstrated for both stroke and ischemic heart disease [41,59,60,61,62,63]. This was confirmed by observations of patients for whom the use of rosuvastatin resulted in a decrease in CRP concentrations with a simultaneous decrease in the risk of myocardial infarction [64,65].

Studies on unstable atherosclerotic plaque in the carotid arteries have shown that their inflammatory activity increases shortly after the ischemic incident and decreases as the atherosclerotic plaque stabilises [66]. Symptomatic atherosclerotic plaque within the carotid arteries may differ in composition, and, thus, in symptomatology, and the nature of plaque damage, e.g., in amaurosis fugax, differs from that observed in transient cerebral ischemia or stroke [67,68]. Therefore, suPAR concentrations may be a potential marker of inflammatory process activity in atherosclerotic plaque and serve as a measure of its stability. Olson et al. reported elevated suPAR levels immediately after the vascular incident in patients with both ischemic stroke and transient cerebral ischemia [69]. By measuring the suPAR concentrations in blood flowing through carotid arteries containing unstable atherosclerotic plaque and comparing them with suPAR concentrations in blood flowing through the radial artery and the antecubital fossa vein, the authors demonstrated that suPAR can be released through the plaque. However, statistical analysis did not determine whether the assessment of suPAR concentrations could provide knowledge about the inflammatory process activity in the plaque or whether such knowledge was restricted to assessing the risk of ischemia development in the course of atherosclerosis. A correlation between suPAR concentrations and age and the coexistence of diabetes was found. In a paper by Elkind et al., the results of studies of patients who underwent internal carotid endarterectomy were analysed [70]. Blood samples collected a day before the procedure showed significantly higher suPAR concentrations in patients with symptomatic plaques than in patients with asymptomatic plaques, but there was no correlation between suPAR concentrations in plaques and plasma levels in the same patient. The suPAR concentration positively correlated with the concentrations of other biomarkers of inflammation, such as high-sensitive CRP (hsCRP) and creatinine, as well as tumour necrosis factor-α (TNF-α), interleukin-1b (IL-1b) or interleukin-6 (IL-6). The suPAR concentration in both atherosclerotic plaque and plasma correlated positively with risk factors such as age, presence of diabetes and female sex. Atherosclerotic plaque damage results in a series of cellular and molecular damage, including lipoproteins, morphotic elements, mechanical vascular damage and inflammatory factors [71]. CRP and suPAR, as acute-phase proteins, are an indicator of the ongoing inflammatory process and highlight the advancement of atherosclerotic plaque [72]. Although they are not stroke-specific markers, such as interleukin 1 (Il-1), interleukin 6 (Il-6) or tumour necrosis factor-α (TNF-α), they suggest the existence of immunological activation in the course of the inflammatory process and tissue damage [58].

Thus, patients with stroke with a thrombotic aetiology, underlying the atherosclerotic plaque process even before the stroke, may show higher concentrations of suPAR and CRP compared to patients with stroke with a cardiovascular aetiology [27]. However, the growth of these markers often takes place within the accepted normal ranges [13]. The gold standard for the evaluation of this process in clinical practice is the determination of high-sensitivity CRP, but the role of suPAR as a more stable molecule, and thus having a greater practical application, is also increasingly indicated [66,73].

Idicula et al. observed that the median CRP concentration on hospital admission was 3 mg/L—but, in some patients, the stroke symptoms increased despite the absence of infection, which was positively correlated with the neurological condition assessed on the NIHSS on admission [74]. High levels of CRP on admission were positively correlated with worse functional status on discharge (measured using the Rankin and Barthel scales) and high mortality rates estimated 2.5 years after stroke. The research also confirmed the positive correlation between suPAR and CRP concentrations and the functional status of patients. Idicula et al. also stated that, in cardiovascular stroke, the CRP concentration after the stroke was higher than that in thrombotic stroke, likely due to higher severity, which leads to higher activation of the immune system [74].

### 4.2. The Role of Lymphocytes

Many authors, in addition to evaluating the usefulness of commonly used markers of inflammation, such as suPAR, CRP and WBC, have highlighted the role of T lymphocytes in immune processes in the course of a stroke. In the analysis by Folyovich et al., it was noted that, in the first hours after a stroke, the number of CD64^+^ lymphocytes suddenly increased and then dropped to lower values than in the control group after seven days [31]. Observing this correlation, the authors suggested that there were two-phase changes in the immune system: the initial activation of the immune system is followed by its suppression, which may result in the development of a secondary infection. The authors assumed that, on the basis of observation of the dynamics of CD64^+^ lymphocyte count changes, it is possible to obtain reliable information about the risk of infection development in the days following stroke. Vogelgesang et al., on the other hand, found that stroke causes an immediate and significant decrease in the number of peripheral CD4^+^ and CD8^+^ lymphocytes [75]. This decrease is most visible in the first 12 h after the stroke, while, in the following days, it gradually returns to normal values. They also showed that normalisation is delayed in patients with follow-up infection. They assumed that the fluctuations in CD4^+^ and CD8^+^ lymphocytes were due to the fact that stroke leads not only to a local inflammatory nervous tissue response reflected in a generalised inflammatory response but also to significant immunosuppression associated with a decrease in peripheral T lymphocytes. This number, determined one day after a stroke, may also be a prognostic factor in the onset of post-stroke infection. In this study, an inverse correlation between the number of T lymphocytes and the NIHSS results was also found. This means that the degree of immunosuppression is reflected in an increasing neurological deficit. The increase in the number of WBC in the work of Vogelgesang et al. was observed at the time of admission, and, in consecutive days of hospitalisation (2nd, 7th and 14th), increased values were observed [75]. In our own observations, an increase in the number of WBC occurred only on the seventh day after stroke, constituting an active parameter but one that is not very useful in the first days after the vascular incident.

Yan et al. observed a significant increase in the number of CD4^+^ lymphocytes immediately after stroke [29]. This lymphocytic line includes regulatory T lymphocytes, which play a role in suppressing the immune response, maintaining immune homeostasis and preventing autoimmunity. It was found that activated T lymphocytes, penetrating through the blood–brain barrier, contribute to secondary damage to nerve tissue in the ischemic zone. It is possible, however, that they also play a protective or regenerative and repairing role within the damaged tissue. This may be due to the presence of cytokines and growth factors supplied by lymphocytes to the lesion site as well as the modulation of microglia activation [76,77].

### 4.3. suPAR and Stroke Risk Factors

In our study, we observed that the coexistence of diseases such as ischemic heart disease, hypertension and type 2 diabetes resulted in higher concentrations of suPAR and CRP in patients in comparison to those not affected by these diseases. In the group of patients with hypercholesterolemia, no differences in suPAR concentration were observed, and higher CRP concentrations were observed only in the patients on the seventh day after stroke. In the patients with atrial fibrillation, no significant differences in the parameters were found. Cigarette smokers differed only in the range of suPAR-3, suPAR-7 and CRP-7. Similar observations were made by many researchers. Haupt et al. found that suPAR levels were elevated in patients with hypertension, diabetes, smoking, alcohol consumption and an unhealthy diet [78]. Higher suPAR concentrations were also found in patients with previous myocardial infarction. On the other hand, the suPAR concentrations among those who quit smoking were comparable to those of patients who never smoked. Concentrations of suPAR were also closely related to biochemical parameters, such as total cholesterol and TG. It was also observed that risk factors such as hypertension, ischemic heart disease, type 2 diabetes and cigarette smoking increase the mortality risk in patients with stroke. These data are consistent with the observations made by Rallidis et al., who observed a significantly higher mortality rate in patients with hypertension and diabetes [79]. However, they did not observe these correlations in patients with ischemic heart disease, hypercholesterolemia, smoking and obesity. The above-mentioned authors also observed that the CRP concentrations were significantly elevated at the time of hospital admission among patients who later died. This parameter was an independent prognostic factor of early mortality-related complications. An increase in CRP concentrations by one unit caused an increase in the risk of early death by 14%.

On the other hand, Persson et al. noted that elevated suPAR concentrations occur in patients with ischemic heart disease and smokers [65]. In their analysis of smokers and non-smokers, they demonstrated that the risk of cardiovascular disease was related to the concentrations of suPAR in both groups. The authors also found a correlation between the concentration of Lp-PLA2 protein and the risk of cardiovascular disease. Concentrations of suPAR were only poorly associated with other markers of inflammation, such as hsCRP and WBC, which is consistent with other publications [52,80].

## 5. Conclusions

To conclude, suPAR can serve as a biomarker of mortality risk in patients with ischemic stroke. The plasma concentration of suPAR increases on the first day following the ischemic stroke and corresponds with the severity of the disease.

## Figures and Tables

**Table 1 jpm-12-00457-t001:** Demographic characteristics.

Risk Factor	Value
Age	70.4 ± 7.9
Gender (F/M)	37/43
Arterial hypertension (%)	56%
Ischemic heart disease (%)	35%
Atrial fibrillation (%)	44%
Type 2 diabetes mellitus (%)	51%
Smoking (%)	23%
Hypercholesterolemia (%)	38%

**Table 2 jpm-12-00457-t002:** suPAR, NIHSS, CRP and WBC values on the first, third and seventh day after ischemic stroke.

Parameter	First Day	Third Day	Seventh Day
suPAR [ng/mL]	6.55 ± 1.66	8.29 ± 3.49	9.16 ± 3.84
NIHSS [pts]	10.71 ± 5.52	12.34 ± 6.42	13.75 ± 8.61
CRP [mg/L]	4.96 ± 2.34	11.76 ± 13.34	17.17 ± 20.13
WBC [thd/mm^3^]	7.32 ± 1.7	9.27 ± 3.57	10.41 ± 3.53

**Table 3 jpm-12-00457-t003:** Risk factors for ischemic stroke and post-stroke mortality.

Death	Yes (*n* = 11)	No (*n* = 69)	*p*
Age	69.5 ± 4.9	70.6 ± 8.3	-
Sex (F/M)	4/7	33/36	-
Arterial hypertension (%)	100	49	*p* < 0.05
Ischemic heart disease (%)	78	59	*p* < 0.05
Atrial fibrillation (%)	45	43	-
Type 2 diabetes mellitus (%)	73	48	*p* < 0.05
Smoking (%)	73	14	*p* < 0.05
Hypercholesterolemia (%)	58	45	-

**Table 4 jpm-12-00457-t004:** Comparison assessed parameters (suPAR, NIHSS, CRP, WBC) and post-stroke mortality.

Death	Yes (*n* = 11)	No (*n* = 69)	*p*
suPAR on day I	8.48	6.24	*p* < 0.05
suPAR on day III	10.74	7.90	*p* < 0.05
suPAR on day VII	12.52	8.62	*p* < 0.05
NIHSS on day I	14.45	10.12	*p* < 0.05
NIHSS on day III	16.00	11.75	*p* < 0.05
NIHSS on day VII	17.36	13.17	*p* < 0.05
CRP on day I	5.91	4.81	-
CRP on day III	9.27	12.16	*p* < 0.05
CRP on day VII	12.68	17.88	*p* < 0.05
WBC on day I	7.40	7.31	-
WBC on day III	8.84	9.34	-
WBC on day VII	10.62	10.38	-

**Table 5 jpm-12-00457-t005:** Comparison of prognostic value of individual parameters.

Parameter	Sensitivity	Specificity	95% CI	Cut-Off Point	Chance Quotient	AUC (0.8–0.95)
suPAR-1 [ng/mL]	90.9	61.8	0.7–0.9	7.64	2.82	0.80
suPAR-3 [ng/mL]	81.8	81.2	0.8–0.9	10.5	8.06	0.89
NIHSS-1 [pkt]	81.8	75.4	0.7–0.9	12.0	2.99	0.83
CRP-3 [mg/L]	81.8	81.5	0.7–0.9	15.6	4.83	0.81
WBC-7 [thd/mm^3^]	72.7	82.4	0.7–0.9	13.7	2.04	0.80

## Data Availability

The data presented in this study are available on request from the corresponding author. The data are not publicly available due to privacy issues.

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
