# Peer review of "Personalised Approach to Diagnosing and Managing Ischemic Stroke with a Plasma-Soluble Urokinase-Type Plasminogen Activator Receptor"

_jpm, 2022, doi:10.3390/jpm12030457_

Round 1

Reviewer 1 Report

The paper entitled: “Personalised approach to diagnose and mange ischemic stroke with plasma-soluble urokinase-type plasminogen activator receptor.”   examines the dynamics of changes in serum levels of suPAR in the ischemic stroke and evaluates the prognostic value of suPAR in determining mortality risk.  This study is original and useful.  Topic is interesting enough to attract the readers’ attention also by virtue of the fact that the frequency of heart attacks in people of all ages is increasing in recent months. I congratulate the authors and I have some curiosity. Therefore, in my opinion, this work could be published after major revision.

My observations are as follows:

  • English revision of the entire manuscript is necessary

  • There are some errors, for example: “with the patient's clinical status and it`s high concentration..” line 34

  • In my opinion some acronyms in the abstract would be better to put them in ful

  • Has the frequency of heart attacks increased in Poland compared to previous years?

  • Exactly in what time frame was the study conducted?

  • Did the authors find differences in demographic characteristics between males and females?

  • Did the authors find differences in the Values of the tested parameters between males and females?

  • Change all ml in mL

  • Explain because the analyses were performed just at First day, Third day and Seventh day

  • Can these parameters that the authors propose as signs of a heart attack be used as predictor markers?
  • In other words, would it be possible to do prevention using these parameters? What could be done to intervene before it is too late?

  • What are the values for subjects with a family history of heart attack?

  • Interestingly, Huttunen et al. reported that in infections caused by Staphylococcus aureus, Streptococcus and Escherichia coli, suPAR levels above 11 ng/ml were a poor prognostic factor. What kind of correlation is there? Microbiome involvement was also found in covid for es. I suggest the authors to read and quote the following works:
  • 3390/medicina57030290
  • 12688/f1000research.77421.1

            and argue on this aspect since also in covid the inflammation factor is relevant

Author Response

Please find attached the response. 

Reviewer 1

The paper entitled: “Personalised approach to diagnose and mange ischemic stroke with plasma-soluble urokinase-type plasminogen activator receptor.”   examines the dynamics of changes in serum levels of suPAR in the ischemic stroke and evaluates the prognostic value of suPAR in determining mortality risk.  This study is original and useful.  Topic is interesting enough to attract the readers’ attention also by virtue of the fact that the frequency of heart attacks in people of all ages is increasing in recent months. I congratulate the authors and I have some curiosity. Therefore, in my opinion, this work could be published after major revision.

My observations are as follows:

  • English revision of the entire manuscript is necessary

We thank the reviewer for this suggestion. We have now revised manuscript with English proofreading.

  • There are some errors, for example: “with the patient's clinical status and it`s high concentration..” line 34

We thank the reviewer for this suggestion. We have now revised manuscript with English proofreading.

  • In my opinion some acronyms in the abstract would be better to put them in ful

              Thank you for this comment. We have reviewed  the abstract accordingly. The only               abbreviation that we use there is CRP, NIHSS and suPAR – taking into consideration the       length and that those abbreviation are commonly used we decided to leave as it it.

  • Has the frequency of heart attacks increased in Poland compared to previous years?

              Thank you for this question. Probably the frequency of heart attacks in Poland increased,              however we are not aware about the study that reported this. Additionally, its not directly    related with our paper therefore we didn't include this information.

  • Exactly in what time frame was the study conducted?

               Thank you for this comment. We have now added this information to the method section.

  • Did the authors find differences in demographic characteristics between males and females?

Thank you for this question. We didn’t find differences between woman and man.

Did the authors find differences in the Values of the tested parameters between males and females?

  • Change all ml in mL

 We thank the reviewer for this comment. We have now adjusted it accordingly.

  • Explain because the analyses were performed just at First day, Third day and Seventh day

The analysis was performed on the first day to have baseline and to reflect the level of activation of immune system. Then 3rd day because it is usually the beginning of stroke induced immunosuppression and 7th day due to  activation of T lymphocytes which highest concentration can be observed on the seventh day after the stroke

  • Can these parameters that the authors propose as signs of a heart attack be used as predictor markers? In other words, would it be possible to do prevention using these parameters? What could be done to intervene before it is too late?

We propose this parameter as a biomarker of ischemia stroke not heart attack. We examined whether the suPAR level dynamic correlates with the mortality risk.

  • What are the values for subjects with a family history of heart attack?

We are not aware about the study which examined the familiar heart attacks in the correlation of suPAR.

  • Interestingly, Huttunen et al. reported that in infections caused by Staphylococcus aureus, Streptococcus and Escherichia coli, suPAR levels above 11 ng/ml were a poor prognostic factor. What kind of correlation is there? Microbiome involvement was also found in covid for es. I suggest the authors to read and quote the following works: 3390/medicina5703029012688/f1000research.77421 and argue on this aspect since also in covid the inflammation factor is relevant

We thank the reviewer for this comment. We agree that this part is not related to the main topic of the paper. Therefore, we decided to delete this part.

Reviewer 2 Report

The authors present a real-world study to assess the dynamics of changes in serum levels of suPAR in the ischemic stroke. The authors show that suPAR is a significant biomarker in the ischemic stroke and mortality risk.

However, there are a few concerns regarding the study design and manuscript writing.

  1. The study only included 80 patients that have ischemic stroke. It is necessary to include comparable number of health controls to assess the changes of suPAR levels.
  2. The inclusion of Type 2 diabetes patients makes the study more complicated, since it may affect the metabolic progress of suPAR levels.

For manuscript wording:

  1. The introduction section is lengthy and the subtitle is unnecessary.
  2. The discussion part includes too many redundant discussions that are not related to results, i.e., the changes of suPAR levels.
  3. The Figure 1 has severe flaw that it lacks y axis label and legends for different curves.

Therefore, I suggest the authors improve the study design by including healthy controls and remove Type 2 diabetes patients. Another concern is that I doubt this study design is a “personalized” approach, since it didn’t use patients’ clinical information to help modeling the prognostic value of suPAR-3.

Author Response

Please find attached the response. 

Reviewer 2

The authors present a real-world study to assess the dynamics of changes in serum levels of suPAR in the ischemic stroke. The authors show that suPAR is a significant biomarker in the ischemic stroke and mortality risk.

However, there are a few concerns regarding the study design and manuscript writing.

  1. The study only included 80 patients that have ischemic stroke. It is necessary to include comparable number of health controls to assess the changes of suPAR levels.

We thank the reviewer for this comment. The study examine the dynamic of change in levels of suPAR in the patients with ischemic stroke. The normal range of suPAR is known therefore we decided that HC will not be included because it will not add for the dynamic of change and assess the suPAR as a biomarker of mortality risk.

Based on our current knowledge we are aware that immunological system is activated by stroke in 30 min. after the incident with further stroke induced immunospression within 3-5 days and then further activation of T lymphocytes which highest concentration that can be observed on the seventh day after the stroke

  1. The inclusion of Type 2 diabetes patients makes the study more complicated, since it may affect the metabolic progress of suPAR levels.

We thank the reviewer for this comment. Taking into consideration that diabetes is one of the major risk factors of ischemic stroke it would be not be beneficial for the study to exclude this group of the patient. Additionally, suPAR levels were examined and established in this subgroup of patients (similarly to smokers etc.) and statistical analysis included this covariant as well. Therefore, we think that stroke biomarker should be examined also in this subgroup pf the patients.

For manuscript wording:

  1. The introduction section is lengthy and the subtitle is unnecessary.

We thank the reviewer for this comment. We have now change it accordingly.

  1. The discussion part includes too many redundant discussions that are not related to results, i.e., the changes of suPAR levels.

We thank the reviewer for this comment. We have now change the discussion accordingly.

  1. The Figure 1 has severe flaw that it lacks y axis label and legends for different curves.

We than the reviewer for this comment we decided to eave this out.

Round 2

Reviewer 1 Report

Accept in present form

Reviewer 2 Report

The authors improves the manuscript by addressing the concerns. 

Minor comment:

The legends and y-axis label in Figure 1 is still lacking, please modify it.